# Mitochondrial Dysfunction and Induction of Apoptosis in Hepatocellular Carcinoma and Cholangiocarcinoma Cell Lines by Thymoquinone

**DOI:** 10.3390/ijms232314669

**Published:** 2022-11-24

**Authors:** Reem J. Abdualmjid, Consolato M. Sergi

**Affiliations:** 1Department of Lab. Medicine and Pathology, University of Alberta, Edmonton, AB T6G 2B7, Canada; 2Anatomic Pathology, Children’s Hospital of Eastern Ontario (CHEO), University of Ottawa, Ottawa, ON K1H 8L1, Canada

**Keywords:** apoptosis, mitochondria, hepatocellular carcinoma, cholangiocellular carcinoma, thymoquinone

## Abstract

Thymoquinone (TQ), a plant-based bioactive constituent derived from the volatile oil of *Nigella sativa*, has been shown to possess some anti-neoplastic activities. The present study aimed to investigate the mitochondria and apoptosis observed when TQ is applied against hepatocellular carcinoma (HepG2) and cholangiocarcinoma (HuCCT1) cells, two of the most common primary tumors of the liver. All cell lines were treated with increasing concentrations of TQ for varying durations. The anti-proliferative effect of TQ was measured using the methoxyphenyl-2-(4-sulfophenyl)-2H-tetrazolium (MTS) assay and resulted in dose- and time-dependent growth inhibition in both cell lines. Cell cycle, apoptosis, and assessment of mitochondria viability by morphology assessment and evaluation of the mitochondrial membrane potential were investigated. The present study confirms that TQ caused cell cycle arrest at different phases and induced apoptosis in both cell lines. A systematic review of rodent animal models was also carried out. Overall, our data seem to represent the most robust results, suggesting that TQ possesses promising therapeutic potential as an anti-tumor agent for the treatment of hepatocellular carcinoma and cholangiocarcinoma.

## 1. Introduction

Hepatocellular carcinoma (HCC) is a highly aggressive and lethal tumor, which arises from the malignant transformation of hepatocytes, and accounts for up to 90% of all hepatic tumors [1,2,3,4,5,6]. The incidence of HCC is increasing around the world, and its prevalence varies according to geographical location and existing risk factors, including hepatitis B (HBV) and hepatitis C (HCV) viruses, among others [6,7,8]. HBV is responsible for 50–80% of HCC cases while HCV for 10–25% of this tumor type [6]. In Asia and African countries, HBV is the predominant virus while HCV is more prevalent in Western countries. The main leading cause of HCC is cirrhosis, which is the result of chronic diffuse hepatitis and is present in 80–90% of all HCC cases. Activation of insulin growth factor (IGF) signaling is a major oncogenic event in several cancers [9,10,11,12]. HCC arises from chronic liver disease of various etiologies, which leads to activation of hepatic inflammation, accumulation of intrahepatic lipid (steatosis), oxidative stress, fibrosis, cirrhosis, and eventual development of autonomous nodules, which accumulate genetic mutations. HCC has been associated with several alterations in cellular signaling pathways and genetic changes [13,14,15,16,17]. The second malignant epithelial tumor of the liver is cholangiocarcinoma (CCA), which is an adenocarcinoma that arises from the epithelial cells of the biliary system with explicit cholangiocyte differentiation [17]. CCA is an aggressive and lethal cancer with a poor prognosis. It is typically diagnosed at advanced stages due to its high potential to infiltrate and metastasize to the liver [18,19]. The incidence of CCA varies globally, with the highest reported prevalence in Thailand, China, Japan, Korea, and Eastern Asia [17,18,19,20,21]. The incidence and mortality of intrahepatic CCA are increasing globally, and this tumor has a low 5-year survival rate following diagnosis of less than 5% [19]. CCA is typically predominant in males usually affected by primary sclerosing cholangitis (PSC), an autoimmune inflammation of the biliary system with marked fibrosis. The other most common predisposing causal factors of CCA are chronic inflammation of the biliary tree, choledochal cysts, and prolonged cholestasis. CCA is also caused by infection with hepatic flukes, including *Opisthorchis viverrini* and *Clonorchis sinensis*, due to the consumption of raw and/or undercooked fish [17,21,22,23]. These parasites mainly accumulate in the bile duct and gallbladder and can persist for years in their hosts, causing chronic inflammation of the biliary system. Other risk factors are chronic hepatolithiasis, exposure to radionuclides, thorotrast (thorium dioxide), dioxin, vinyl chloride, radon, asbestos, nitrosamine, and dichloropropane as most recently identified other than common liver viral infections [24,25,26,27,28]. The step-by-step molecular mechanisms involved in CCA pathogenesis are not fully understood. However, some of the proposed molecular pathways include autonomous or self-sufficiency proliferation, escape from senescence, resistance to apoptosis, and unlimited replication [16,17,19,29]. 

Chemotherapeutic agents, which kill rapidly dividing neoplastic cells, are commonly used in cancer therapy, but highly proliferative normal cells, as seen in teenagers or youth, are also affected [2]. Doxorubicin and cisplatin have been used as cancer therapy. However, doxorubicin has limitations because of its side effects on the heart, liver, and kidney, although the use of matrix metalloproteinases seems to be promising [30]. Cisplatin similarly exhibits several adverse effects on the kidney, neural tissue, and liver. Vulnerability to viral infections due to immunosuppression, drug resistance due to the enhancement of the drug efflux pump, activation of drug detoxification mechanisms, and stimulation of DNA damage repair are some of the chemotherapy drawbacks. Although some botanicals are hepatotoxic [31], there has been much interest in the last decade in investigating the anti-cancer potential of phytochemical compounds extracted from natural sources [32].

*Nigella sativa* (*N. sativa*) is an annual herbaceous flowering plant that belongs to the *Ranunculaceae* family. It is native to countries bordering the Mediterranean Sea and southwestern Asian countries. In English-speaking countries, *N. sativa* is also known as black or “blessed” seeds, black cumin, black caraway, nutmeg flower, and Roman coriander [31,33,34]. Morphologically, *N. sativa* is a small shrub with divided tapering green leaves and delicate rosaceous flowers, usually white, yellow, or purple. It has large capsulated fruit that contains numerous tiny black seeds (Figure 1a). Since the ancient Egyptians and Greeks, *N. sativa* has been known for centuries but has also been referenced in several religious and ancient texts, particularly in Islamic and Jewish traditions [35,36]. For thousands of years *N. sativa* seeds have been traditionally used in cooking as a food preservative and as a spice in cheese, bread, and soups [37,38,39]. Seeds and oils of *N. sativa* have been well known for their therapeutic potential to promote and maintain health and treat several diseases and conditions [39,40,41]. Black seeds have been shown to have a variety of chemical constituents [33]. The fixed oil accounts for 37% of seeds’ constituents, including unsaturated fatty acids, while volatile oil accounts for 0.4–2.5% and contains many other constituents, of which thymoquinone (TQ) is the most abundant one [40,42,43] (Table 1). 

TQ has been demonstrated to exert numerous biological activities, including anti-inflammatory, anti-bacterial, anti-viral, anti-diabetic, and immunomodulatory activities [44,45,46,47,48,49]. The molecular structure of TQ consists of a benzene ring conjugated with para-substituted dione, with a methyl side chain in position 2 and an isopropyl side chain in position 5 (Figure 1b). The anti-neoplastic and antioxidant potential of TQ have been intensely investigated in in vivo and in vitro models [50,51,52]. An anti-proliferative effect of TQ was found in several cancers, including osteosarcoma [53], breast and ovarian adenocarcinoma [54], colorectal carcinoma [55], hepatocellular carcinoma [56], pancreatic adenocarcinoma [57], neoplastic keratinocytes [57,58], fibrosarcoma and lung carcinoma [59], glioma/glioblastoma [60], prostatic carcinoma [59,61], and myeloblastic leukemia [62].

Despite several studies investigating the anti-tumor activities of TQ on different types of cancers, the anti-tumor effects of TQ on HCC and CCA cells have not been extensively studied [63,64]. Moreover, controversial results are present in the literature [65]. In the present study, we hypothesized that TQ, the primary active constituent of *N. sativa* seed extract, induces apoptosis in HCC and CCA cell lines. To test this hypothesis, we aimed to (1) evaluate the anti-proliferative and cytotoxic activities of TQ in HCC and CCA cell lines, (2) investigate the growth inhibitory effect of TQ, (3) determine the mode of death of tumor cells treated with TQ (apoptosis or necrosis), and (4) determine the subcellular alterations induced by TQ that lead to tumor cell death.

## 2. Results

### 2.1. TQ Inhibits the Proliferation of Tumor Cell Lines

The anti-proliferative effect of TQ on HCC and CCA cell lines and normal THLE-3 cell lines following treatment with different concentrations of TQ was measured using the colorimetric MTS assay (Figure 1c–e). All cell lines were exposed to varying concentrations of TQ (10–200 µM) for differing durations (12, 24, and 48 h). Negative controls (cells not exposed to TQ) and vehicle controls (cells treated with 0.05% DMSO) were included. No positive controls were included in this experiment. Concentration- and time-related inhibition of cell growth was observed in TQ-treated cells compared to non-treated controls.

### 2.2. Effect of TQ on Cell Cycle Distribution

Cell cycle analysis of HepG2 and HuCCT1 is shown in Figure 2a (HepG2) and Figure 2b (HuCCT1). To determine whether the cytotoxic effect of TQ is associated with disruption of the cell cycle, changes in the cell cycle progression of the HepG2 and HuCCT1 cell lines were investigated. Cells were treated with increasing doses of TQ for 24 h, followed by PI staining of the DNA content. Flow cytometry was employed to quantify the cell populations in different cell cycle phases (sub-G1, G1, S, and G2/M phases). The distributions of the cell cycle phases in the HepG2 cell lines and HuCCT1 are shown in Figure 3a–d, respectively. Table 2 highlights the cell cycle distribution of the cell lines following exposure to the TQ treatments. The results of the sub-G1 assay, which is one of the most widely used assays to determine apoptosis by flow cytometry estimating the fractional DNA content, are also shown.

In HepG2 cells, a significant reduction in the percentage of cell growth was observed at 25 µM TQ and higher after 12 h of incubation (*p* < 0.05, Figure 3a,b). At 24 h, the percentage of cell viability after treatment with 25 mM TQ and higher was statistically significantly lower than the control. At 48 h, a significant increase in cell viability was observed at 10 and 25 µM while concentrations from 50 µM and higher resulted in significantly decreased cell viability. In the HuCCT1 cell lines, cell proliferation was significantly reduced at 75 µM TQ and higher after 12 h of incubation (*p* < 0.05, Figure 3c,d). At 24 h, 50 µM TQ and higher resulted in a significant decrease in viable cells. Incubation for 48 h at 25 µM and higher also resulted in substantial suppression of growth. In both cell lines, DMSO (0.05%) treatment did not result in observable cytotoxic effects.

The IC_50_ values of the HepG2 and HuCCT1 cell lines for the respective durations of exposure to TQ are shown in Figure 4a–c (HepG2) and Figure 4d–f (HuCCT1) and listed in Table 3. The differences in these values indicate that TQ cytotoxicity might be related to the type of cancer and could be cell line specific. Since substantial variability in the cell viability was observed at the 12-h incubation time point, an incubation period of 24 h was chosen for further experiments, representing the shortest time when cells exhibited a complete biochemical response to TQ. The IC_50_ values for TQ were 100 µM or less. Therefore, 10–100 µM concentrations of TQ were selected for the subsequent studies. TQ did not exert a cytotoxic effect on standard THLE-3 cell lines, which are SV40-immortalized non-neoplastic hepatocytes.

Treatment of the HepG2 cell lines with TQ did not significantly accumulate cells in the cell cycle phases G1, S, or G2/M. The number of cells in the G1 and S phases decreased with higher doses of TQ. The percentage of cells in the G2/M phase increased after incubation with 10, 25, and 50 mM, and decreased after incubation with 75 and 100 mM TQ. 

In the HepG2 cells (Figure 5a) and HuCCT1 cells (Figure 5b), the percentage of apoptotic cells increased by increasing the TQ concentration (*p* < 0.05, Figure 5a,b). This data clearly shows that cells in the G1 phase increased dramatically in a dose-dependent manner compared to the non-treated control. The number of cells in the G1 phase increased from 46.8% under the control conditions to 60.3% after incubation with 100 mM TQ. A significant decrease in the G2/M phase was observed (*p* < 0.05), but no significant changes were observed in the S phase. The results shown are one representative of three individual experiments. Values are presented as the mean of three individual experiments ± SEM. Flow cytometry plots are shown in Figure 5c for HepG2 and Figure 5d for HuCCT1. For both cell lines, the percentage of cells in the sub-G1 phase, representing the apoptotic population, was significantly increased in a dose-related manner (*p* < 0.05). The figure shows the detection of apoptosis in both (A) HepG2 cell lines and (B) HuCCT1 cell lines using annexin V/propidium iodide (PI) analysis. The annexin V/PI assay is used to identify the mode of cell death induced by TQ and distinguish whether death is due to apoptosis or necrosis. Annexin V is often used to detect apoptotic cells due to its ability to bind to phosphatidylserine. This phospholipid is a marker of apoptosis when it is on the outer leaflet of the plasma membrane. PI is an impermeant membrane dye that is commonly excluded from viable cells. It binds to double-stranded DNA by intercalating between base pairs. Values are presented as the mean of three individual experiments ± SEM and TQ promote the apoptosis of tumor cells. 

The IC_50_ values are converted to the pIC_50_ scale. Considering the minus sign, higher values of pIC_50_ indicate exponentially more potent inhibitors. The values of IC_50_ in both cell lines are quite similar. Additionally, pIC_50_ is usually given in terms of the molar concentration (mol/L, or M), thus requiring IC_50_ in units of M.

As described above, HepG2 and HuCCT1 cell lines were treated with increasing concentrations of TQ for 24 h followed by Annexin V/PI staining. The representative dot plots of flow cytometric analyses of HepG2 and HuCCT1 cells demonstrated four different distributions: live or healthy cells (low left; annexin V and PI negative), cells in early apoptosis (low right; annexin V positive and PI negative), cells in late apoptosis (upper right; annexin V and PI positive), and dead or necrotic cells (upper left; annexin V negative and PI positive). The results indicate that treating cells with TQ resulted in a significant, dose-related increase in the percentage of apoptotic cells compared to non-treated controls.

Additionally, a slight increase in the number of necrotic cells was seen. The percentage of live cells was significantly reduced. The apoptotic-inducing effect of TQ was demonstrated by the shift in the HepG2 and HuCCT1 cells toward the lower and upper right quadrants (Figure 5c,d). In Figure 6, which shows the distribution of the flow cytometric analysis of FITC annexin V/PI staining of the (A) HepG2 and (B) HuCCT1 cell lines, values are presented as the mean of three individual experiments. The total number of early and late apoptotic cells is higher compared to the number of necrotic cells, indicating that apoptosis is the primary mechanism by which TQ causes cell death in both cell lines (Figure 6).

### 2.3. Determination of Morphological Changes in Cells Treated with TQ

Confocal microscopy was used to assess the morphological alterations of HepG2, HuCCT1 and standard THLE-3 cell lines following exposure to different concentrations of TQ for 24 h. the HepG2 and HuCCT1 cell lines treated with TQ showed a noticeable reduction in the cell number, which was more evident at higher concentrations of TQ (Figure 7). Cells also began to lose cell–cell contact. They exhibited shrinkage, and they were seen to round up. Some cells were clustered together and detached from the surface of the culture dishes. Some cells showed chromatin condensation and membrane blebbing. These observed changes point to apoptosis-mediated cell death. As the TQ concentration increased, more effects were observed in the cells, such as the formation of apoptotic bodies with fragmented nuclei.

In contrast, untreated cells and vehicle control cells treated with DMSO displayed no observable effects. They were well spread and attached to the surface of the culture dish, with prominent and intact nuclei. Similarly, no effect on the standard HEK293T cell lines was observed after TQ treatment, as confirmed by light microscopic examination. The morphologic alterations in HepG2 cells (a), HuCCT1 cells (b), THLE-3 cells (c), and HEK293T cells (d) shown by confocal microscopy are illustrated in Figure 7.

Thus, it proves that TQ had no toxicity on HEK293T cell lines, although the standard THLE-3 cell line displayed slight changes in its morphological appearance in response to TQ exposure. The THLE-3 cell lines are SV40 virus sequence immortalized human liver epithelial cells that have demonstrated some slow growth in culture. Therefore, we used HEK293T cells as an additional morphological control. HEK293 is an embryonic kidney immortalized cell line, which shows an epithelial morphology and remarkable good growth in cell culture. HEK293 cells have been initially referred to as 293tsA1609neo and contain SV40. TQ disrupts the mitochondrial membrane potential (MMP).

Mitochondria play an essential role in cellular apoptosis. Hence, the changes in the mitochondrial membrane potential (MMP) of TQ-treated cells were detected using the fluorescent cationic JC-1 dye. In the normal polarized mitochondrial membrane, the JC-1 dye emits red fluorescence (dimer), and in the depolarized mitochondrial membrane, it emits green fluorescence (monomer). Thus, MMP is an essential index of mitochondrial function and depends on the extent of mitochondrial impairment.

The exposure of cell lines to TQ resulted in decreased MMP in a dose-dependent manner in both the HuCCT1 and HepG2 cell lines compared to non-treated controls. In addition, due to the altered MMP in TQ-treated cells, the JC-1 dye did not accumulate in the mitochondria and persisted in its monomeric form, emitting green fluorescence.

A gradual reduction in MMP was observed until it was remarkably altered at 100 μM for HepG2 cells and 100 μM for HuCCT1 cells (Figure 8, Figure 9, Figure 10 and Figure 11). In the untreated controls, the JC-1 dye aggregated in the mitochondria, emitting a bright red fluorescence. Figure 8 shows the disruption of MMP in HepG2 cells following exposure to the TQ concentration for 24 h. Figure 9 shows the rationale of using JC-1 to determine the depolarization/disruption of the mitochondrial membrane, with three color histograms of the red, green, and blue fluorescence of 0-TQ, DMSO, and 50-μM TQ for HepG2 cells, and the mitochondrial depolarization for HepG2 using JC-1. Figure 10 highlights the disruption of MMP in HuCCT1 cells following exposure to the TQ concentration for 24 h and Figure 11 shows the mitochondrial depolarization using JC-1 for HuCCT1 using image analysis.

### 2.4. TQ Protective Effect Is Trustworthy

We conducted a systematic review of animal models that use TQ to protect several organs from tissue damage by applying several chemical compounds. After a review of 59 papers (mouse) and 130 papers (rat), we accumulated 183 articles and extracted 13 articles, which was conducted according to the Guidance of Good Clinical Practice (IARC, WHO). There are numerous reports suggesting some preservation of the tissue integrity in the setting of TQ administration [37,66,67,68,69,70,71,72,73,74,75,76,77,78,79,80,81,82,83,84,85,86,87,88,89,90,91,92]. Table 4 summarizes the rodent models targeting the therapeutic potential of TQ [62,63,64,65,66,67,68,69,70,71,72,73,92].

## 3. Discussion

In this study, we report on the anti-proliferative effect of TQ, which caused cell cycle arrest at different phases and induced apoptosis in malignant liver cell lines. We confirm previous results with a robust methodology and suggest that TQ possesses promising therapeutic potential as an anti-tumor agent for the treatment of hepatocellular carcinoma and cholangiocarcinoma.

Most existing anti-cancer drugs have been developed to target a specific signaling pathway responsible for tumorigenesis [78]. Natural compounds derived from plants, herbs, and spices have garnered substantial interest as an alternative therapy for treating cancer patients. These are frequently accessible and have been shown to target multiple signaling pathways with minimal or no side effects [79]. The cytotoxic effect of TQ has also been reported in multi-drug resistant (MDR) cell lines. TQ suppressed growth and induced apoptotic cell death in cisplatin-resistant canine osteosarcoma cell lines (COS31/rCDDP) [54]. TQ also exerted a significant anti-tumor effect against MDR uterine sarcoma, leukemia, and pancreatic cancer cell lines [57]. Furthermore, TQ causes little to no cytotoxicity in non-cancerous cell lines [59,60,80]. It seems that the anti-proliferative effects of TQ are due to the induction of apoptosis and cell cycle arrest [51,54,81]. TQ has been reported to induce apoptosis by either p53-dependent or p53-independent pathways. Indeed, one study highlighted the apoptotic effect of TQ in HCT-116 colon cancer cells via upregulation of p53 and p21, leading to a decrease in anti-apoptotic Bcl-2 protein expression (p53-dependent pathway) [55].

Conversely, another study reported the induction of apoptosis in HL-60 myeloblastic leukemia cells by disrupting the mitochondrial membrane potential and activation of caspase-8 (p53-independent pathway) [62]. in addition to p53, other molecular targets have been reported to be involved in TQ-induced anti-cancer and apoptotic effects. It has been reported that TQ-induced apoptotic effects in MCF-7/DOX breast cancer cells are due to the upregulation of PTEN with suppression of p-Akt [82]. Additionally, TQ was found to upregulate p73 with a decrease in UHRF1 in p53 mutant acute lymphoblastic leukemia Jurkat cells [83]. TQ also caused inhibition of STAT3 and NF-kB and their regulatory gene products in U266 multiple myeloma cells and KBM-5 human myeloid cells, respectively [84,85]. TQ also induced apoptosis in acute lymphoblastic leukemia Jurkat cells through the production of reactive oxygen species (ROS), resulting in a loss of the mitochondrial membrane potential [83]. Nevertheless, activation of the JNK and p38 MAPK pathways, with downregulation of mucin-4, was induced by TQ and led to apoptosis in FG/COLO357 pancreatic cancer cells [86]. TQ has also been reported to suppress growth by arresting cells in different phases of the cell cycle in various tumors. 

The chemotherapeutic potential of TQ has been investigated in some cancer animal models [41,51]. TQ also exerted protection against toxicities caused by these conventional chemotherapies without affecting their therapeutic efficacy in conjunction with other clinically available anti-cancer drugs. TQ reduced the size and number of aberrant crypt foci (ACF) in mice with colon cancer induced by 1,2-dimethylhydrazine [87] and inhibited the incidence of benzo(a)pyrene (BP)-induced forestomach tumors in Swiss albino mice [88]. Administration of TQ to rats with N-nitroso diethylamine (NDEA)-induced hepatic cancer reduced liver injury and tumor markers, inhibited the formation of hepatic nodules, and diminished tumor diversity [89]. TQ enhanced the efficacy of ifosfamide and resulted in lower mortality and decreased body weight loss in Ehrlich ascites carcinoma-bearing mice [90]. Similar anti-tumor effects were found in mice harboring fibrosarcoma induced by 20-methylcholanthrene [91]. TQ also enhanced the anti-tumor effect of cisplatin in mice and rats and averted cisplatin-induced nephrotoxicity [92]. TQ also improved the efficacy of doxorubicin and prevented cardiotoxicity caused by doxorubicin, and further studies are on the way to evaluating this compound in our doxorubicin animal model [30,93,94]. Another study demonstrated a protective effect of TQ against carbon tetrachloride hepatotoxicity in mice [95]. Finally, TQ enhanced the anti-tumor efficacy of gemcitabine and oxaliplatin in a pancreatic cancer model [96]. The growth inhibitory effect of TQ has also been reported in other xenograft models of C4-2B prostatic cancer cells [59], NCI-H460 lung cancer cells [97], HPAC pancreatic cancer cells [96], and HCT-116 colon cancer cells [87].

In this work, TQ was investigated for its conceivable anti-cancer activity on HCC and CCA cell lines. Our results demonstrate that TQ inhibits the proliferation of both cancer cell lines and induced cell cycle disruption and apoptosis through the mitochondrial pathway. The anti-proliferative activity of TQ was tested using the colorimetric MTS assay. The findings showed a concentration- and time-dependent reduction in the cell viability of TQ-treated cells compared to non-treated controls. We consolidated that TQ is cytotoxic to both HepG2, with IC_50_ values of 50.74, 34.23, and 50.84 µM for 12, 24, and 48 h, respectively, and HuCCT1 cell lines, with IC_50_ values of 155.1, 63.12, and 61.59 µM for 12, 24, and 48 h, respectively. The HepG2 cell lines were shown to be more sensitive to TQ treatment. These results support the findings of a previous study that reported significant inhibition of the growth of HepG2 cell lines corresponding to TQ exposure in a concentration-dependent manner [98,99], although the historical IC_50_ value of TQ after 24 h was reported to be 350 µM. A similar study approaching our IC_50_ value was recently reported [100]. 

Additionally, TQ was reported to reduce the cell viability of two HCC cell lines, HepG2 and Huh-7, in a dose-dependent manner [101]. Only one study has substantially suggested an anti-proliferative effect of TQ on HuCCT1 cells [102]. Our results strongly indicate that TQ causes a concentration- and time-dependent reduction in the number of viable cells. The anti-proliferative effect of TQ on standard THLE-3 cell lines was also tested in this study. The results indicate that lower concentrations of TQ exerted no influence on healthy cells. The IC_50_ after 24 h of exposure to TQ was 200 µM, which is the highest concentration used in the Xu et al. study [102]. Our results indicate that TQ has a limited cytotoxic effect on non-cancerous cell lines, which correlates with the findings of most of the studies. 

Conversely, TQ has been reported to have some cytotoxic effect on standard human cell lines, including lung fibroblast cells (IMR90) [60], intestinal cells (FHs74Int) [103], and prostate epithelial cells (BPH-1) [59]. One limitation of these experiments is that no positive control was used, such as using a cytotoxic drug (saponin) or lytic detergent. However, according to previously published studies on TQ and the manufacturer’s protocol, positive controls are not required. We are aware that in vivo experiments and the expression of mitochondrial pathway-related proteins are necessary to verify the pharmacological mechanism of TQ. However, our robust data confirm the TQ effect on cancer cell lines.

Further studies will target the p53-dependent and non-p53-dependent pathways and our doxorubicin C57BL/6J mouse model [30]. Although other studies have targeted TQ, these findings confirm previous results and highlight the robustness of the chosen methods. To the best of our knowledge, this is the most comprehensive study on TQ’s effect on these liver cancer cell lines.

Cell cycle arrest checkpoints provide possible targets for therapeutic agents to treat cancers, and quinones are reported to be one of the drugs that cause cell cycle arrest [104,105,106]. Accordingly, the capacity of TQ to induce cell cycle arrest in HCC and CCA cell lines was investigated. HepG2 cell lines did not show an accumulation of cells in any cell cycle phase. However, the number of cells in the G2/M phase increased with lower concentrations (10, 25, and 50 µM) and decreased with 75 and 100 µM. The decrease in cells in the G2/M phase at high concentrations may be due to cell loss during cellular preparations, with only 10,000 events available. If 20,000 events had been captured, the percentage of cells would have increased to 75 and 100 µM TQ, confirming the cell cycle arrest of HepG2 cell lines in the G2/M phase. This result corresponds to findings reported in a similar study, where TQ was found to arrest HepG2 cells in the G2/M phase of the cell cycle [101].

In contrast to the observations of our investigation, another study reported that the exposure of HepG2 cells to TQ induced cell cycle arrest in the G1 phase [107]. These variations in the ability of TQ to arrest HepG2 cells in the G1 and G2/M phases could be due to the different concentrations of TQ used in each study and different duration periods. Another possibility is the type of solvent used to dissolve TQ. In the first study carried out by Ashour et al. (2014) [101], TQ was prepared in ethanol. The TQ concentrations were 6, 12.5, 25, 50, and 100 µM. Cells were exposed to TQ for 6, 12, and 18 h. In Hassan et al.’s study (2008) [107], the researchers dissolved TQ in methanol, and the concentrations (25, 50, 100, 200, and 400 µM) were prepared in the medium. However, in that study, the 350 µM TQ concentration was the only dose tested for cell cycle analysis with incubation periods of 6 and 12 h. Further studies have investigated the cell cycle arrest activity of TQ in a similar way to our methodology. They have demonstrated the G2/M arrest in other cell lines, including I7 mouse spindle carcinoma cells via an increase in p53 and decrease in cyclin B [58], MNNG/HOS human osteosarcoma cells due to p21^WAF1^ upregulation [53], MCF-7/DOX doxorubicin-resistant breast cancer cells through an increase in p53 and p21 proteins [108], and MNNG/HOS osteosarcoma cell lines by the upregulation of p21^Waf1^ [53].

However, there is controversy in the literature because the results of the previous studies indicate that TQ induces cell cycle arrest in the G1 phase in HuCCT1 cell lines, made evident by the increased G1 population. These findings oppose a previous study, where TQ caused G2/M arrest in HuCCT1 cells [102]. In this study, TQ was used at concentrations of 20–60 µM, and cells were treated for 48 h. In our experiments, the TQ concentrations were 10–100 µM, and the incubation period was 24 h. The prolonged exposure to TQ for 48 h may explain the G2/M arrest seen in Xu et al.’s study [102].

Nevertheless, several studies have demonstrated the G1-arresting property of TQ in different cancer cell lines. Indeed, TQ was shown to induce G1 arrest in HCT 116 human colorectal carcinoma cells through the regulation of p53 [55], acute lymphoblastic leukemia Jurkat cell lines via a p73-dependent pathway as evident by p73 siRNA [83], LNCaP prostatic cancer cells by an increase in p21 and p27 and a decrease in E2 transcription factor [59], SP-1 mouse papilloma carcinoma cells through an increase in p16 and decrease in cyclin D1 [58], COS31 canine osteosarcoma cell lines [54], and MDA-MB-468 and MDA-MB-231 triple-negative breast carcinoma cells with mutant p53 [109]. Overall, the results of the current and previous studies undeniably confirm the ability of TQ to cause a cell-line-dependent cell cycle arrest in different phases, resulting in suppression of growth. Moreover, in both cell lines studied in the present study, TQ-mediated cell cycle arrest was associated with an increase in the apoptotic cell population (Sub-G1). The mechanism involved in TQ-induced cell cycle disruption in cancer cells has been reported to be a result of TQ on CDKs, their inhibitors, and cyclins. The primary mechanism of TQ-induced cell cycle arrest seems to be an upregulation of p53 and p21. Additionally, TQ targets other proteins such as p16, p73, and cyclin B, determining a cell cycle arrest.

In addition to anti-proliferative and cell cycle arrest effects, the apoptotic effects of TQ were investigated. We found that TQ treatments of HepG2 and HuCCT1 cell lines increased the percentage of cells in the early and late stages of apoptosis in a concentration-dependent manner. These findings suggest that TQ-mediated cell death occurs via induction of apoptosis in these cell lines, consistent with data from previous studies showing the apoptotic effect of TQ in several cancer cell lines. Similar studies investigating the effect of TQ on HepG2 cells have reported that TQ increased the percentage of cells in early and late apoptosis in a concentration-dependent manner [101]. Another study described that the treatment of HepG2 cells with TQ caused 57% of cells to undergo apoptosis [107]. Additionally, the presented apoptotic effect of TQ on HuCCT1 was consistent with a previous study reporting the apoptosis-inducing ability of TQ in HuCCT1 cell lines [102].

These results agree with earlier studies that tested the apoptotic activity of TQ on various cancer cell lines. For example, one study showed that 100 µM TQ induces apoptosis in HCT-116 colorectal carcinoma cells [55]. Moreover, TQ was found to mediate apoptosis in p53-null HL-60 cells in a dose- and time-dependent manner [62]. The induction of apoptosis was also seen in TQ-treated SP-1 and I7 cells [55]. Furthermore, the apoptotic effect of TQ was reported in primary effusion lymphoma (PEL) cell lines, including BC-1, BC-3, BCBL-1, and HBL-6, where TQ caused dose-dependent apoptosis [110].

TQ-mediated apoptosis has been observed in other cancer cells, including DLD-1 colon cancer cell lines [103], human umbilical vein endothelial cells (HUVECs) [111], NCI-H460 and NCI-H146 lung cancer cells [97], M059J, ACHN human renal cell carcinoma [112], and M059K human glioblastoma cell lines [60]. The induction of apoptosis in different cancer cells following treatment of TQ was found to be mediated by p53-dependent and p53-independent pathways [55,62,112]. Additionally, the p73, NF-κB, STAT3, and PTEN pathways were also involved in TQ-mediated apoptosis in different cell models [52]. Finally, oxidative stress via the generation of ROS may mediate TQ-induced apoptosis [103]. Prolonged exposure to TQ was found to induce necrosis in cancerous cell lines. Indeed, TQ induced 43% necrosis in COS31 canine osteosarcoma cells [54]. This aspect suggests that TQ induces apoptosis in most cancer cells, but with prolonged exposure and higher TQ concentrations, necrosis was also observed. It is evident that the mode of cell death after TQ treatment varies depending on the cancerous cell line [113,114].

The induction of apoptosis in HepG2 and HuCCT1 cell lines was further supported by cellular morphological changes following exposure to increasing concentrations of TQ for 24 h. The results indicated that TQ treatment with TQ caused significant alterations in the cellular morphology compared to non-treated controls. After 24 h of treatment with TQ, HepG2 and HuCCT1 cells displayed noticeable changes and distress such as cell shrinkage, reduced cytoplasm, and reduced nuclear size. Moreover, TQ treatment caused cells to round up, reduce adhesion, and detach from the surface of the plates. The cell number reduction was more prominent with higher TQ concentrations, and most cells were separated and aggregated together, forming clusters. In addition, most cells were floating in media, showing fragmentation and the presence of background debris (apoptotic bodies). These features are hallmarks and signs of apoptosis [115]. It is not surprising that some initial response to TQ seems paradoxical, but it has been discussed as part of a more complex phenomenon called hormesis or hormesis-like status. Hormesis regards adaptive reactions of biological systems to moderate environmental or self-imposed challenges. The physical system may improve its functionality and/or tolerance to somewhat more severe challenges [116,117].

In contrast, untreated control cells in all cell lines exhibited a high proliferation rate, with around 90% confluence. Durable adhesion to the surface of the culture dish was observed. The morphological assessment of standard HEK293T cell lines was performed using light microscopy. TQ-treated cells remained similar in shape to untreated control cells. The proliferation rate of cells was consistent over the concentration ranges of TQ employed in this study. These findings confirmed that TQ has no cytotoxic effect on non-cancerous cell lines. Alterations in the morphology of TQ-treated standard THLE-3 cell lines were also examined using confocal microscopy. The results revealed some changes in the cellular appearance of cells because of the TQ treatment. These results confirm the significant cytotoxic effect of TQ on cancerous cell lines and suggest its possible impact to induce cell death in these cell lines through an apoptotic pathway.

We found an alteration of MMP in cells treated with TQ. The effect of TQ on MMP of HepG2 and HuCCT1 cells was investigated to determine whether mitochondrial disruption mediated TQ-induced apoptosis. Cells were treated with different doses of TQ for 24 h. Following treatment, cells were stained with JC-1 stain to permit quantification of the changes in MMP and visualized using fluorescent microscopy. TQ-treated HepG2 and HuCCT1 cell lines exhibited a loss of MMP in a dose-related manner, evidenced by the reduction of red fluorescence, indicating depolarized mitochondria.

Conversely, untreated controls showed a marked red fluorescent, indicative of intact and polarized mitochondria. These findings suggest that the TQ-induced apoptosis in HepG2 and HuCCT1 cell lines was mediated via the mitochondrial pathway, as demonstrated by the dissipation of MMP. This discovery agrees with previous studies on TQ. For example, the apoptotic effect of TQ on MDA-MB-231 and MDA-MB-468 breast cancer cells was reported to be induced by the mitochondrial pathway, shown by the loss of mitochondrial membrane integrity, release of cytochrome c, and cleavage of PARP [118]. Another study by El-Mahdy et al. (2005) [62] reported that TQ, a p53-mediated-independent pathway, induced apoptosis in HL-60 myeloblastic leukemia, as evident by the disruption of MMP and activation of caspases, including caspase 8, 9, and 3.Pl

Moreover, it investigated the apoptotic effect of TQ on HPAC human pancreatic cancer cells, showing that TQ-induced apoptosis in HPAC was mitochondrial in origin as indicated by the dose-dependent release of cytochrome c, although this publication was recently retracted [96]. However, the contribution of mitochondria to apoptosis induction in MG63 osteosarcoma cell lines, as indicated by the activation of caspase 9 and caspase 3, remains coherent [53]. Figure 12 schematizes the effects of TQ. We cartooned the effects of TQ on a five-spiked star, highlighting the inhibitory effects against cancer and the protective effect against diabetic nephropathy and retinopathy (Figure 12).

Usually, a positive control group is a group that is not directly exposed to the experimental compound but that is exposed to some other compounds, which are known to produce the expected effect. This kind of control is particularly useful to validate the experimental procedure. Our experimental design did not contemplate positive controls with chemical compounds that are ordinarily efficacious and/or effective in oncology. This is a limitation of this study, but we are targeting the next phase of experiments, including several commercially available drugs, and we will compare TQ with other drugs that are currently used in oncology.

The data presented in this study improve the understanding and knowledge of the possible mechanism responsible for the anti-proliferation effect of TQ against hepatic cancer cell lines. The main objective of this research project was to investigate the anti-cancer activity of commercially available TQ against hepatic cancer cell lines. To test the hypothesis that TQ would exhibit anti-cancer effects on HCC and CCA, TQ was investigated for potential anti-proliferative and cytotoxic effects using a colorimetric MTS assay. Cell cycle disruption was investigated using flow cytometric analysis. At the same time, apoptosis was targeted using an annexin V/PI detection kit, and microscopic examination of the morphological changes and their effect on MMP were assessed using JC-1 dye. In light of the results demonstrated here, TQ was found to inhibit proliferation and induce cytotoxicity toward HepG2 and HuCCT1 cell lines in a dose- and time-dependent manner.

Additionally, TQ was shown to induce cell cycle disruption and caused cell cycle arrest in the G2/M and G1 phases of HepG2 and HuCCT1. TQ also exhibited apoptotic effects against these cell lines, as shown by the increased number of cells in the early and late stages of apoptosis and visualized features of apoptosis under microscopy, including cell shrinkage, chromatin condensation, membrane blebbing, and the presence of apoptotic bodies. Finally, TQ caused the loss of MMP in a dose-dependent manner, confirming TQ-induced apoptosis was mediated via the mitochondrial pathway. Investigation of the molecular mechanisms by which TQ induces cell death is essential to elucidate its potential application to treat hepatic cancers fully. 

Recently, Jehan et al. identified that TQ selectively promotes HCC apoptosis in synergy with the p53 status, which may harbor relevance for most current tumor RNA modification studies [119,120]. In this study, the cytotoxic effects of TQ, alone or in combination with cisplatin and doxorubicin, were investigated in HCC cells. Hep3B cells with a p53-null status were more sensitive to TQ than HepG2 and SMMC-7721 cells harboring wild-type p53. Moreover, shRNA-mediated p53 silencing in HCC cells dramatically enhanced TQ-induced apoptosis. It seems that TQ harmoniously improves the anti-cancer activity of cisplatin and doxorubicin. In addition, the loss of p53 sensitizes HCC cells to TQ-induced apoptosis.

Overall, our work provides scientific evidence of the anti-tumor activity of TQ. It may further consolidate the concept of using this compound as an alternative or complementary therapy for treating HCC and CCA. Assessment of the safety, efficacy, and dosage of TQ in vivo is warranted and determination of the time-dependent influence of TQ. In the future, evaluation of the effects of TQ in xenograft and transgenic animal models of liver cancer is essential to resolve the interaction between normal and tumor cells to determine the optimum dosage and delivery method of the drug to treat tumors.

## 4. Materials and Methods

### 4.1. Cell Lines and Culture Conditions

HCC HepG2 cells were purchased from the American Type Cultural Collection (ATCC) (Manassas, VA, USA) and were grown in Dulbecco’s Modified Eagle’s Medium (DMEM) (Invitrogen Canada Inc., Burlington, ON, Canada). The cells were supplemented with 10% fetal bovine serum (FBS) (PAA Laboratories Inc., Etobicoke, ON, Canada) and 1 mL gentamicin (50 µg/mL) (Gibco^TM^, ThermoFisher Scientific, Missisauga, ON, Canada). CCA HuCCT1 cells were obtained from the cell culture bank of the Japan Health Sciences Foundation. They were grown in Roswell Park Memorial Institute medium (RPMI-1680, Invitrogen Canada Inc., Burlington, ON, Canada) and supplemented with 10% FBS and 1 mL gentamicin (50 µg/mL). Immortalized normal human liver THLE-3 cells were purchased from the American Type Cultural Collection (ATCC) (Manassas, VA, USA) and cultured in previously coated flasks. The cells were supplemented with 0.01 mg/mL fibronectin, 0.03 mg/mL bovine collagen type 1, and 0.01 mg/mL bovine serum albumin dissolved in bronchial epithelial cell growth medium (BEGM) for 24 h. The cells were cultivated in BEGM supplemented with a BEGM SingleQuot kit except for GA-1000 and epinephrine (Lonza Group Ltd., Basel, Switzerland), 10% FBS, and 1 mL gentamicin (50 µg/mL). The human embryonic kidney HEK293T cells were a gift from Dr. Judith Hugh, University of Alberta, Canada, and were maintained in Dulbecco’s modified eagle’s medium (DMEM, Invitrogen Canada Inc., Burlington, ON, Canada) supplemented with 10% FBS and 5 mL penicillin/streptomycin (100 U/mL) (PAA Laboratories). All cells were grown in a monolayer culture in their proper media and maintained in a humidified atmosphere in a 5% CO_2_ incubator at 37 °C. They were passaged when they reached 70–90% confluency using trypsin plus 0.25% EDTA for 5 min in a CO_2_ incubator. The Ethics Committee of the University of Alberta approved this study (Pro00020718, study title: Perturbance of Mitochondrial Function and Induction of Apoptosis in Liver Tumor Cell Lines by Seed Extract Fractions of *Nigella sativa*).

### 4.2. TQ Preparation

A 1 M stock solution of TQ was prepared by dissolving 0.1642 g TQ powder in cell-culture tested, sterile 1 mL 100% dimethyl sulfoxide (DMSO) (Sigma-Aldrich, St. Louis, MO, USA). A 2× stock solution (20–400 µM) was prepared in 1 mL 100% DMSO. All stocks were divided into aliquots, covered with aluminum foil, and stored at −20 °C until use. TQ was prepared at different concentrations (10–200 µM) by diluting this compound in fresh cell culture medium before use to avoid deposits. The final concentration of DMSO was 0.05%. DMSO is an aprotic solvent that can promote cell fusion and cell differentiation and increase the permeability of lipid membranes. 

### 4.3. Cell Proliferation Assay

A commercially available colorimetric kit, CellTiter 96 AQueous One Solution Cell Proliferation Kit (MTS) (Promega, Madison, WI, USA), was used to determine the number of viable cells. This method is based on the bio-reduction of the MTS tetrazolium compound by NADPH or NADH produced by dehydrogenase enzymes of active cells into a purple Formazan product, which can be detected by measuring the absorbance at 490 nm. The amount of Formazan produced is directly proportional to some metabolically active cells. The kit was used according to the manufacturer’s protocol. Briefly, HepG2, HuCCT1, and THLE-3 cell lines were seeded at a density of 6000 cells/well in quadruplicate wells of 96-well tissue culture plates with 50 µL complete media. Cells were allowed to attach and grow overnight at 37 °C in a 5% CO_2_ humidified atmosphere. The next day, cell media were replaced by 50 µL complete fresh media containing variable concentrations of TQ (10–200 µM) and cultured in a humidified incubator for 12, 24, and 48 h. A colorimetric 3-(4,5-dimethyl-thiazol-2yl)-5-(3-carboxy-methoxyphenyl)-2-(4-sulfophenyl)-2H-tetrazolium (MTS) reagent was added (20 µL/well) into each well and incubated for 4 h. The absorbance at 490 nm was recorded using a Spectra Max M3 Microplate Reader (Molecular Devices, San Jose, CA, USA). The background absorbance of the sample wells was normalized to that of the negative control wells, which only contained media. The percentage of cell proliferation was calculated using the following formula:


$$\%\mathrm{Proliferation}=\frac{\mathrm{Absorbance}\;\mathrm{of}\;\mathrm{treated}\;\mathrm{wells}}{\mathrm{Absorbance}\;\mathrm{of}\;\mathrm{control}\;\mathrm{wells}}\times 100$$


Three independent experiments in quadruplicate were carried out for the HepG2 and HuCCT1 cell lines, and only one individual experiment in quadruplicate was carried out for the THLE-3 cell lines. From the obtained MTS results, the IC_50_ value, which is defined as the concentration of TQ that causes 50% cell death in these cell lines, was calculated for each concentration by plotting a dose–response curve.

### 4.4. Cell Cycle Analysis

To determine the distribution of cell lines in each phase of the cell cycle, propidium iodide (PI) (Sigma-Aldrich, St. Louis, MO, USA) was used to stain the DNA content of each cell line. At a density of 1 × 10^6^–3 × 10^6^ cells/dish, the HuCCT1 and HepG2 cell lines were seeded in 30-mm tissue culture plates in 5 mL complete medium. Cells were incubated and allowed to adhere in the CO_2_ atmosphere. After 24 h of adherence, cells were incubated with different TQ concentrations (10−100 µM) obtained by dissolving TQ in 5 mL complete fresh medium and incubated for 24 h. After that, both adherent and non-adherent cells were collected and centrifuged at 300× *g* for 5 min. The cell pellets were washed twice with fetal calf serum (FCS) washing buffer and fixed with 70% ice-cold ethanol for a minimum of 24 h at −20 °C. Fixed cells were centrifuged at a higher speed and washed with FCS washing buffer twice. Cell pellets were stained with 1 mL PI solution, which was obtained by combining 50 µg/mL PI and 3.8 mM sodium citrate in PBS with the addition of 50 μL 10 µg/mL RNase A, mixed well, and placed in 4 °C protected from light until use. The distribution of PI-stained cells across the cell cycle was analyzed by a fluorescence-activated cell sorter (BD FACSCalibur™, Becton, Dickinson and Company, Mississauga, ON, Canada) flow cytometer, and the data were processed using CellQuest Pro 6.0 software (Becton, Dickinson and Company, Mississauga, ON, Canada). Data were collected from three individual experiments.

### 4.5. Apoptosis and Cell Necrosis Assay

An annexin V-FITC apoptosis detection kit (BD Biosciences, Becton, Dickinson and Company, Mississauga, ON, Canada) was used to quantify the percentage of cells undergoing apoptosis and determine the mode of cell death, whether by apoptosis or necrosis, in the presence or absence of TQ. The experiment was carried out according to the manufacturer’s protocol. Briefly, cells were seeded (1× 10^6^–3 × 10^6^) per dish and allowed to adhere overnight in the CO_2_ incubator. Following 24 h of incubation, TQ treatments (10–100 µM) were added, and plates were incubated for another 24 h in the CO_2_ atmosphere. Both adherent and non-adherent cells were trypsinized, collected, and centrifuged for 5 min at 300× *g*. Next, cell pellets were washed with 2 mL cold PBS twice, resuspended in 100 µL 1× binding buffer, and stained with 5 µL FITC Annexin V and 5 µL PI for 15 min in the dark at room temperature. Following incubation, 1 mL 1× binding buffer was added, and the analysis was carried out using a BD FACS Calibur flow cytometer within 1 h. Data was collected from three individual experiments.

### 4.6. Morphological Assessment

Morphological changes in the different cell lines after treatment with increasing concentrations of TQ were optically recorded. Briefly, HepG2, HuCCT1, and THLE-3 cell lines (4000 cells/well) were plated in 8-well glass chambered slides in 200 µL complete medium and left to grow overnight in the CO_2_ incubator. Then, 200 µL full fresh medium containing the TQ treatments (10–200 µM) was added to each well and incubated for 24 h. HepG2, HuCCT1, and THLE-3 cells were visualized using the Zeiss AxioObserver Z1 (inverted) confocal microscopy (Zeiss 710 LSM, Carl Zeiss Canada Ltd., Toronto, ON, Canada). HEK293T cell lines were seeded at a density of 0.3 × 10^6^ cells/well in 6-well tissue culture plates and allowed to adhere overnight. Cells were then treated with TQ for 24 h and visualized using digital microscopy. All images were processed using ZEN 2009 software (Carl Zeiss Canada Ltd., Toronto, ON, Canada).

### 4.7. Mitochondrial Membrane Potential

Changes in the mitochondrial membrane potential (MMP or ΔΨm) were detected using the fluorescent JC-1 (5, 5′, 6, 6′-tetrachloro-1,1′,3,3′-tetraethyl-benzimidazolyl-carbocyanine iodide) dye (Molecular Probes, Invitrogen, Germany), which fluoresces red/orange fluorescence in intact and healthy mitochondria and fluoresces green fluorescence in the cytoplasm in disrupted mitochondria. Between 10,000 and 15,000 cells/well of HepG2 and HuCCT1 cell lines were cultured in 8-well glass chambered slides and incubated in a CO_2_ incubator for 24 h. Increasing concentrations of TQ (10–200 µM) were added to each well and incubated in a CO_2_ incubator for 24 h. JC-1 stock solution (1 mg/mL) was prepared in DMSO. The JC-1 working solution at a final concentration of 0.1 µg/mL was prepared in a complete medium, and 40 µL was added to each well and incubated for 30 min in the CO_2_ incubator. Cells were visualized using the Zeiss AxioObserver Z1 (inverted) confocal microscopy (Zeiss 710 LSM). All images were processed using ZEN 2009 software (Carl Zeiss Canada Ltd., Toronto, ON, Canada). The main role of measuring the mitochondrial transmembrane potential (ΔΨ_M_) is to understand the cell dynamics. ΔΨ_M_ drives adenosintriphosphate synthesis (ATP) using oxidative phosphorylation. An increase in ATP generation or mitochondrial dysfunction can promote a decrease in ΔΨ_M_. JC-1, a cationic carbocyanine dye (green), shows potential-dependent accumulation in mitochondria where it begins aggregate (J aggregates) and upon depolarization, JC-1 remains as a monomer, exhibiting green fluorescence. Thus, in intact mitochondria, JC-1 aggregates, giving a red signal while unhealthy mitochondria evidence JC-1 monomers, which are seen as a green signal in the cytosol. Color histograms were evaluated using Fiji software program (NIH, Bethesda, MD, USA). A decrease in the red (~590 nm)/green (~529 nm) fluorescence intensity ratio by exposure to TQ is indicative of depolarization/disruption of the mitochondrial membrane. 

### 4.8. Statistical Analysis

All obtained data were presented as mean ± S.E.M. of three or four replicates from three individual experiments. All data were analyzed statistically using Statistical Package for Social Sciences (SPSS) software (version 21.0) (IBM, Armonk, NY, USA). The statistical analysis for the anti-proliferative assay was conducted using a one-way analysis of variance (ANOVA) test to determine the significant differences between different concentrations, followed by Dunnett’s post hoc test to compare the significant differences among concentrations in comparison to non-treated controls. For the cell cycle and apoptotic assays, the non-parametric Kruskal–Wallis test was used to determine significant changes across the different groups. The α-level for significance was set at 5%.

### 4.9. Systematic Review

A systematic review was carried out to identify the animal models using either mouse or rat (rodents) to support a protective effect of this compound against potential carcinogens or immunological dysregulations. We searched SCOPUS, Medline, PubMed, and Google Scholar for items including “mouse” or “rat” and “thymoquinone”. 

## Figures and Tables

**Figure 1 ijms-23-14669-f001:**
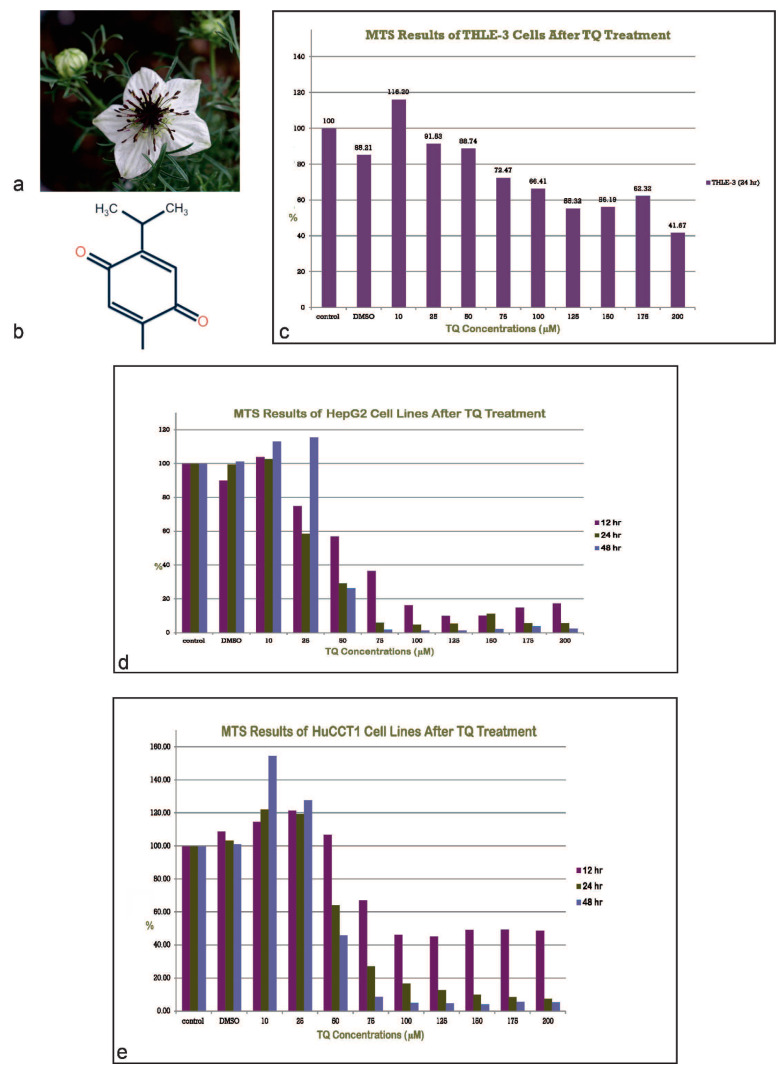
(**a**) *Nigella sativa* flowering plant. (**b**) Chemical structure of thymoquinone. (**c**) Anti-proliferative effect of TQ on THLE-3 cell lines. There is minimal anti-proliferation using methoxyphenyl-2-(4-sulfophenyl)-2H-tetrazolium (MTS). (**d**) Anti-proliferative effect of TQ on HepG2 cell lines. There is evidence of an anti-proliferative effect of TQ as measured using the methoxyphenyl-2-(4-sulfophenyl)-2H-tetrazolium (MTS assay), which resulted in dose- and time-dependent growth inhibition in the cell line. (**e**) Anti-proliferative effect of TQ on HuCCT1 cell lines. There is evidence of an anti-proliferative effect of TQ as measured using the methoxyphenyl-2-(4-sulfophenyl)-2H-tetrazolium (MTS) assay, which resulted in dose- and time-dependent growth inhibition in the cell line (h, hr: hour).

**Figure 2 ijms-23-14669-f002:**
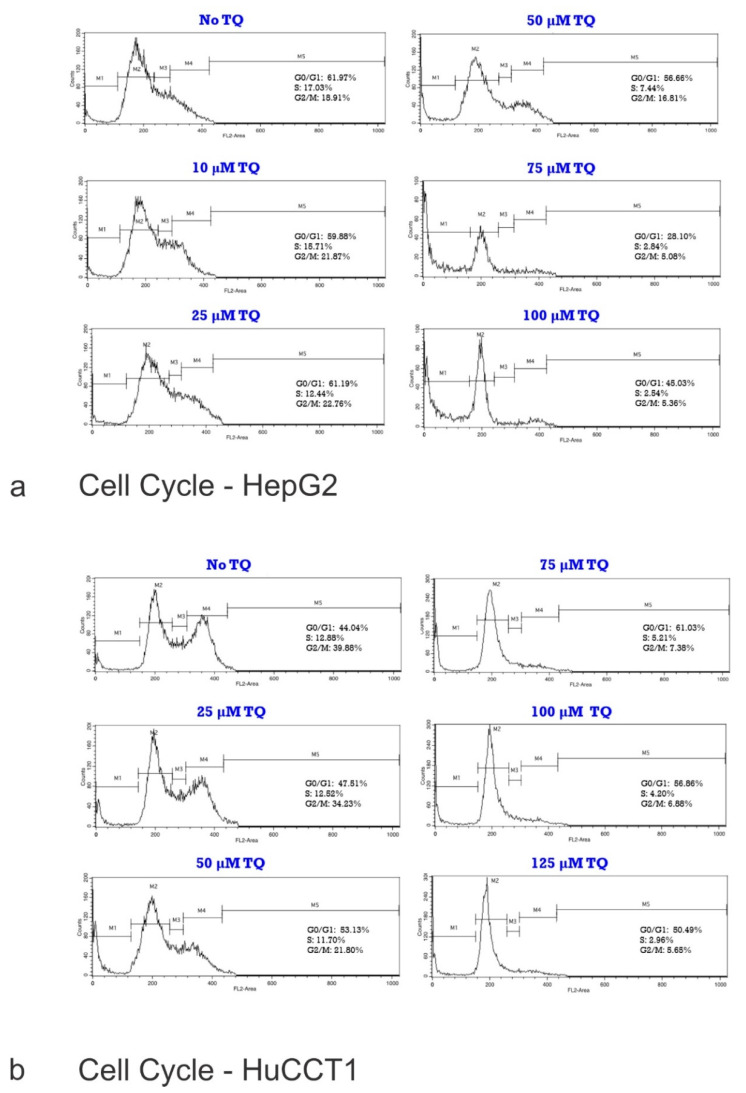
(**a**) Distribution of Cell Cycle Phases in HepG2 cell lines. To determine whether the cytotoxic effect of TQ is associated with disruption of the cell cycle, changes in cell cycle progression of the HepG2 cell line were investigated. Cells were treated with increasing doses of TQ for 24 h, followed by PI staining of the DNA content. Flow cytometry was employed to quantify the cell populations in the different cell cycle phases (sub-G1, G1, S, and G2/M phases). The flattening and re-shaping of the curve is remarkable as the concentration of TQ increases (see text for details). (**b**) Distribution of the Cell Cycle Phases in the HuCCT1 cell lines. To determine whether the cytotoxic effect of TQ is associated with disruption of the cell cycle, changes in the cell cycle progression of the HuCCT1 cell line were investigated. Cells were treated with increasing doses of TQ for 24 h, followed by PI staining of the DNA content. Flow cytometry was employed to quantify the cell populations in the different cell cycle phases (sub-G1, G1, S, and G2/M phases). The flattening and re-shaping of the curve is remarkable as the concentration of TQ increases (see text for details).

**Figure 3 ijms-23-14669-f003:**
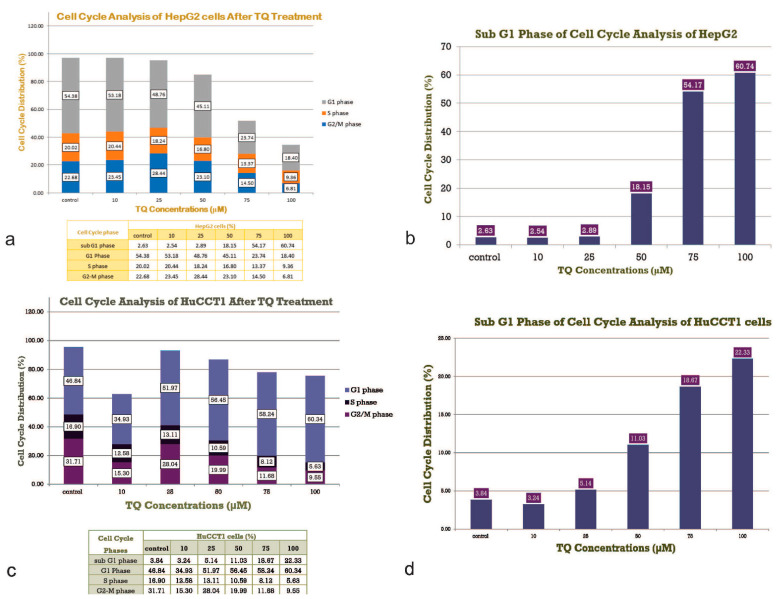
(**a**) Cell cycle Analysis of HepG2 cells after TQ Treatment. (**b**) Histogram of the Cell Cycle with the Sub G1 Phase of the Cell Cycle of the HepG2 cell lines. (**c**) Cell cycle Analysis of HuCCT1 cells after TQ Treatment. (**d**) Histogram of the Cell Cycle with the Sub G1 Phase of the Cell Cycle of the HuCCT1 cell lines. In (**a**,**b**), a histogram of the Cell Cycle distribution of the HepG2 cell lines is shown. The results shown are one representative of three individual experiments. In (**c**,**d**), the distribution of the cell cycle phases in the HuCCT1 cell lines is shown. Values are presented as the mean of 3 individual experiments ± SEM.

**Figure 4 ijms-23-14669-f004:**
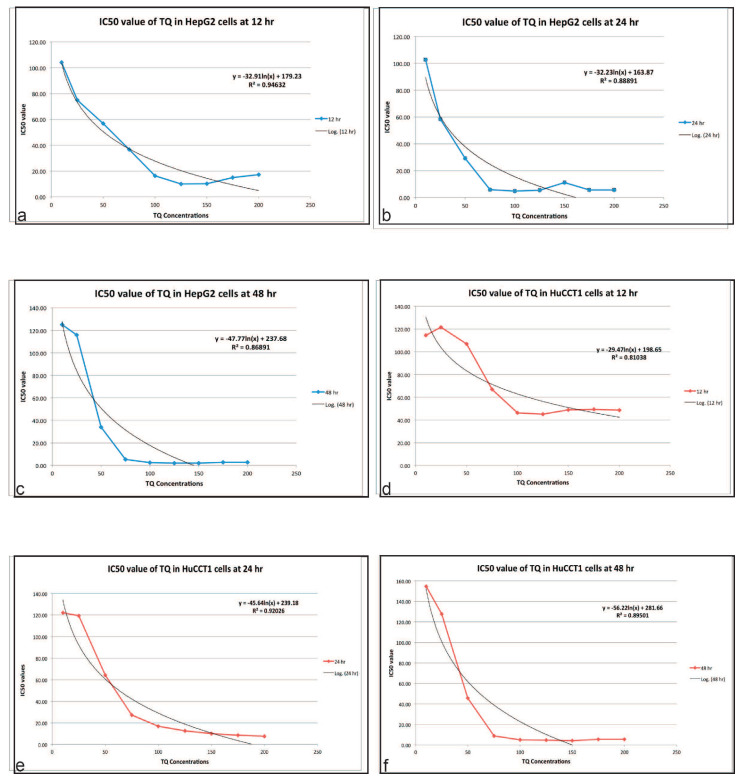
(**a**–**c**) IC_50_ Values of TQ in HepG2 cells at 12, 24, and 48 h. (**d**–**f**) IC50 Values of TQ in HuCCT1 cells at 12, 24, and 48 h. IC_50_ is a quantitative measure that indicates how much of a particular inhibitory substance is needed to inhibit, in vitro, a given biological process or biological component by 50%. IC_50_ values are converted to the pIC_50_ scale. Considering the minus sign, higher values of pIC_50_ indicate exponentially more potent inhibitors. The values of IC_50_ in both cell lines are quite similar. Additionally, pIC_50_ is usually given in terms of the molar concentration (mol/L, or M), thus requiring IC_50_ in units of M (h, hr: hour).

**Figure 5 ijms-23-14669-f005:**
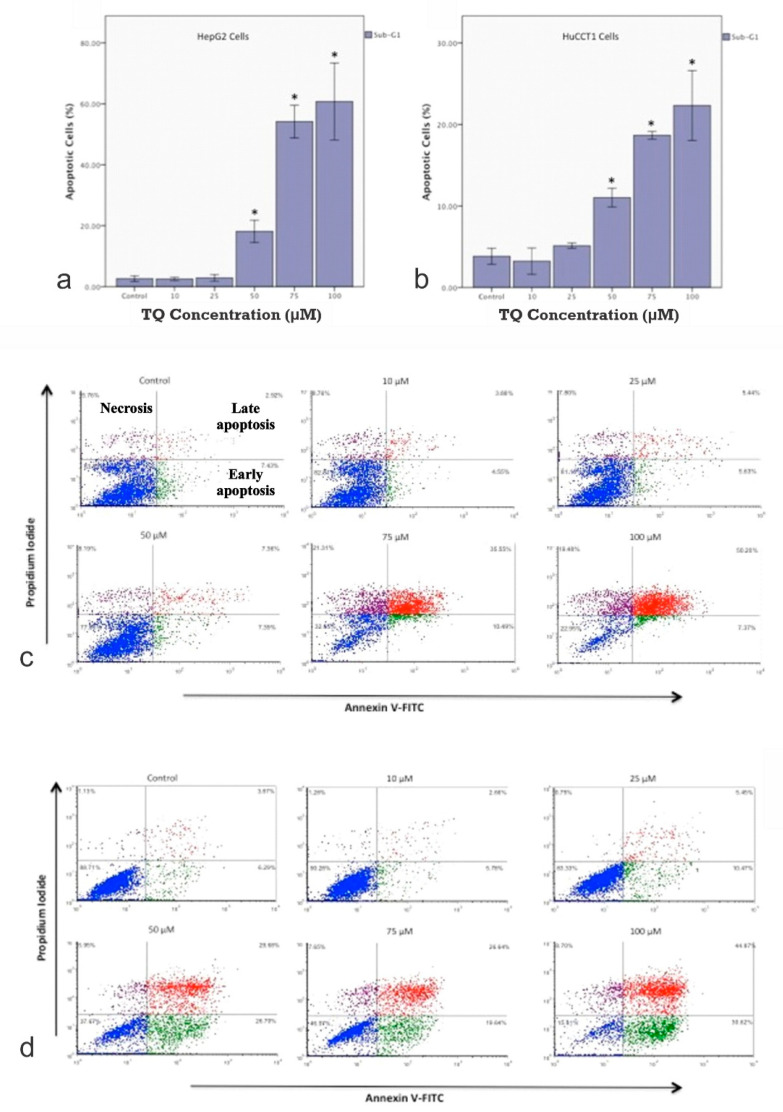
(**a**,**b**) Histogram of the distribution of Apoptotic Cells after TQ Treatment. (**c**) Flow cytometric plots of FITC Annexin V/PI staining of HepG2 cell lines. (**d**) Flow cytometric plots of FITC Annexin V/PI staining of HuCCT1 cell lines. Here, the flow cytometric plots of FITC Annexin V/PI staining of (A) HepG2 and (B) HuCCT1 cell lines (experiments performed in triplicate) are shown. Annexin V is a Ca2+-dependent phospholipid-binding protein, which can bind to phosphatidylserine (PS) protein that externalizes from the inner membrane to the outer surface of the plasma membrane of cells undergoing apoptosis. Annexin V is used in combination with PI dye to distinguish early from late apoptotic cells. Live cells have intact plasma membranes that exclude PI, whereas the membranes of damaged cells are permeable to PI (*: *p* < 0.05).

**Figure 6 ijms-23-14669-f006:**
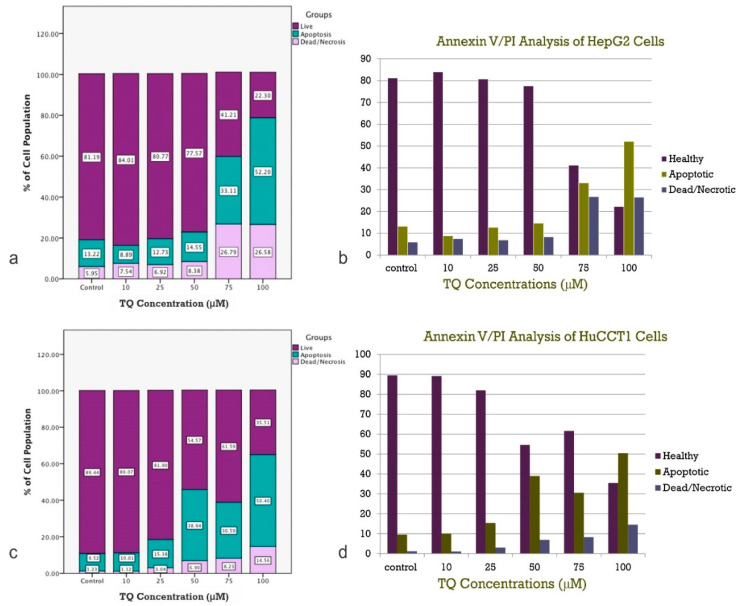
(**a**,**b**) Flow cytometric Analysis of FITC using Annexin V/PI staining of HepG2 cell lines (see text for details). (**c**,**d**) Flow cytometric Analysis of FITC using Annexin V/PI staining of HuCCT1 cell lines (see text for details).

**Figure 7 ijms-23-14669-f007:**
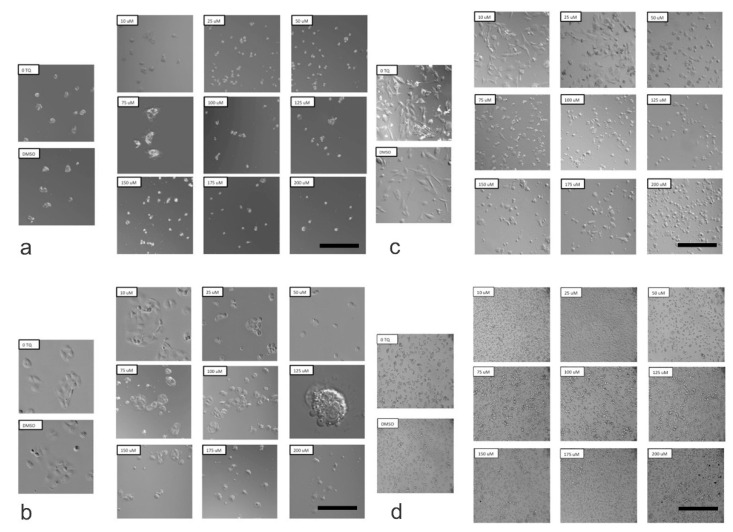
Morphological Alterations Associated with TQ treatments in (**a**) HepG2 cells, (**b**) HuCCT1 cells, (**c**) THLE-3 cells, and (**d**) HEK293T cells by confocal microscopy (scale bar: 50 μm). Confocal microscopy was used to assess the morphological alterations of the HepG2, HuCCT1, and standard THLE-3 cell lines following exposure to different concentrations of TQ for 24 h. HepG2 and HuCCT1 cell lines treated with TQ showed a noticeable reduction in the cell number, which was more evident at higher concentrations of TQ. However, the morphology was little or not altered in the SV40-immortalized hepatocytes (THLE-3) and SV40-immortalized renal epithelial cells (HEK293T). The best representative images of each group are shown. (**a**) comprises up and down of the figure on the left are 0-TQ and DMSO, respectively, and the nine microphotographs from left to right: on the first row, 10, 25, and 50 μM TQ; on the second row, 75, 100, and 125 μM TQ; and on the third row, 150, 175, and 200 μM TQ (scale bar: 50 μm). (**b**) comprises up and down of the figure on the left are 0-TQ and DMSO, respectively, and the nine microphotographs from left to right: on the first row, 10, 25, and 50 μM TQ; on the second row, 75, 100, and 125 μM TQ; and on the third row, 150, 175, and 200 μM TQ (scale bar: 50 μm). (**c**) comprises up and down of the figure on the left are 0-TQ and DMSO, respectively, and the nine microphotographs from left to right: on the first row, 10, 25, and 50 μM TQ; on the second row, 75, 100, and 125 μM TQ; and on the third row 150, 175, and 200 μM TQ (scale bar: 50 μm). (**d**) comprises up and down of the figure on the left are 0-TQ and DMSO, respectively, and the nine microphotographs from left to right: on the first row, 10, 25, and 50 μM TQ; on the second row, 75, 100, and 125 μM TQ; and on the third row, 150, 175, and 200 μM TQ (scale bar: 50 μm).

**Figure 8 ijms-23-14669-f008:**
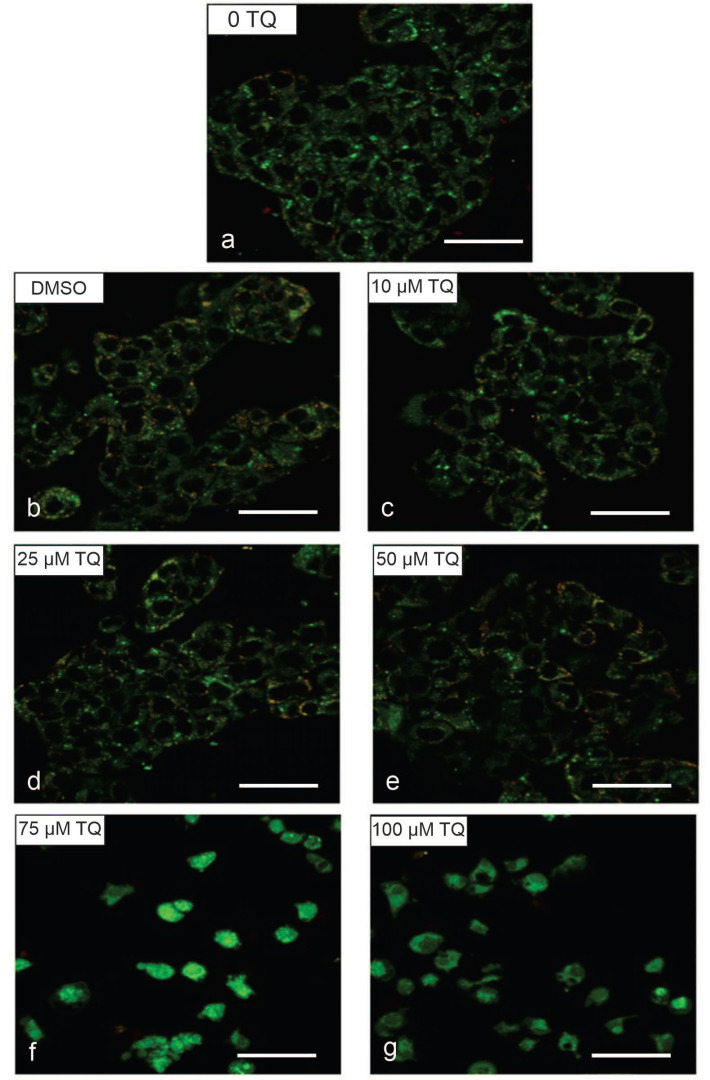
MMP disruption in HepG2 cells following exposure to TQ for 24 h (scale bar: 50 μm). MMP was evaluated using JC-1 by image techniques in the cancer cells after being exposed to 0–100 μM TQ: 0 μM TQ (**a**) and DMSO (**b**), and 10 (**c**), 25 (**d**), 50 (**e**), 75 (**f**), and 100 μM (**g**) TQ. The best representative images of each group are shown.

**Figure 9 ijms-23-14669-f009:**
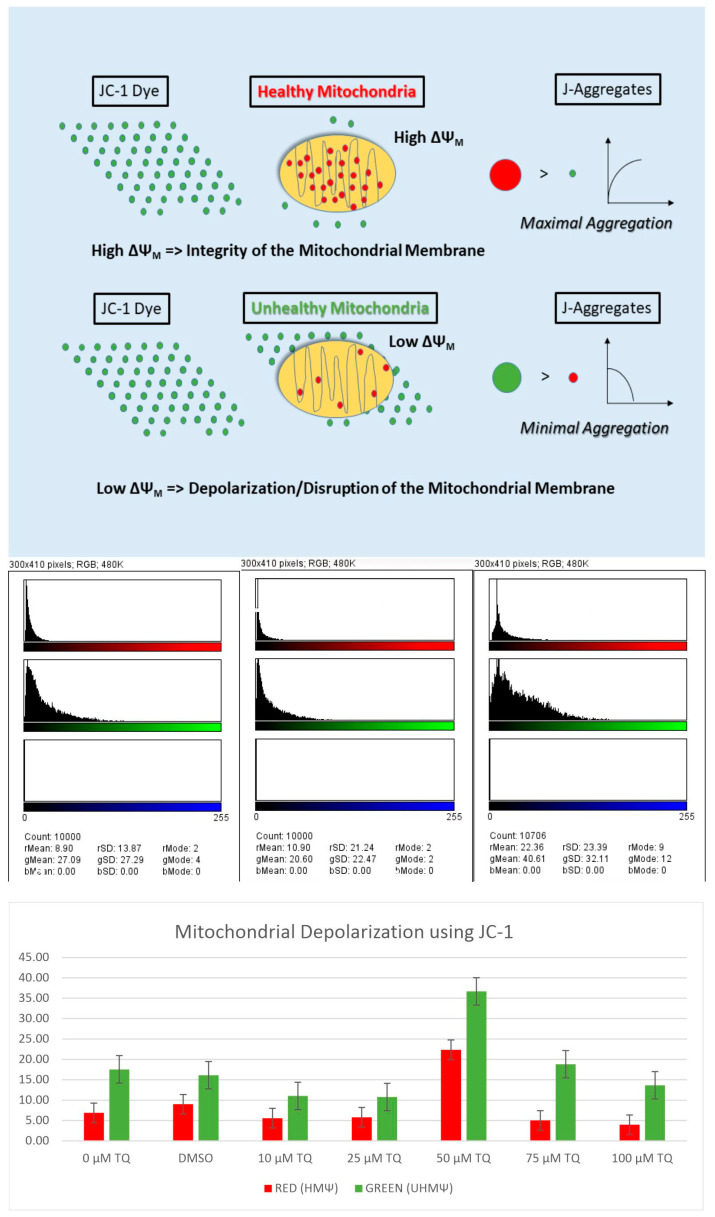
The upper panel shows the rationale of using JC-1 to determine the depolarization/disruption of the mitochondrial membrane, with the three-color histograms of red, green, and blue fluorescence representing 0 TQ, DMSO, and 50 μM TQ for HepG2 cells, and the mitochondrial depolarization for HepG2 using JC-1. A decrease in the red (~590 nm)/green (~529 nm) fluorescence intensity ratio after exposure to TQ is indicative of depolarization/disruption of the mitochondrial membrane. Three independent experiments were carried out for each concentration. The variation in the mitochondrial depolarization is significant (*p* < 0.001, two-tailed test). Red means healthy mitochondria (HMΨ) while green means unhealthy mitochondria (UHMΨ) (RGB, red, green, and blue fluorescence channels; for each fluorescence channel evaluation, there is mean, the standard deviation, and the mode).

**Figure 10 ijms-23-14669-f010:**
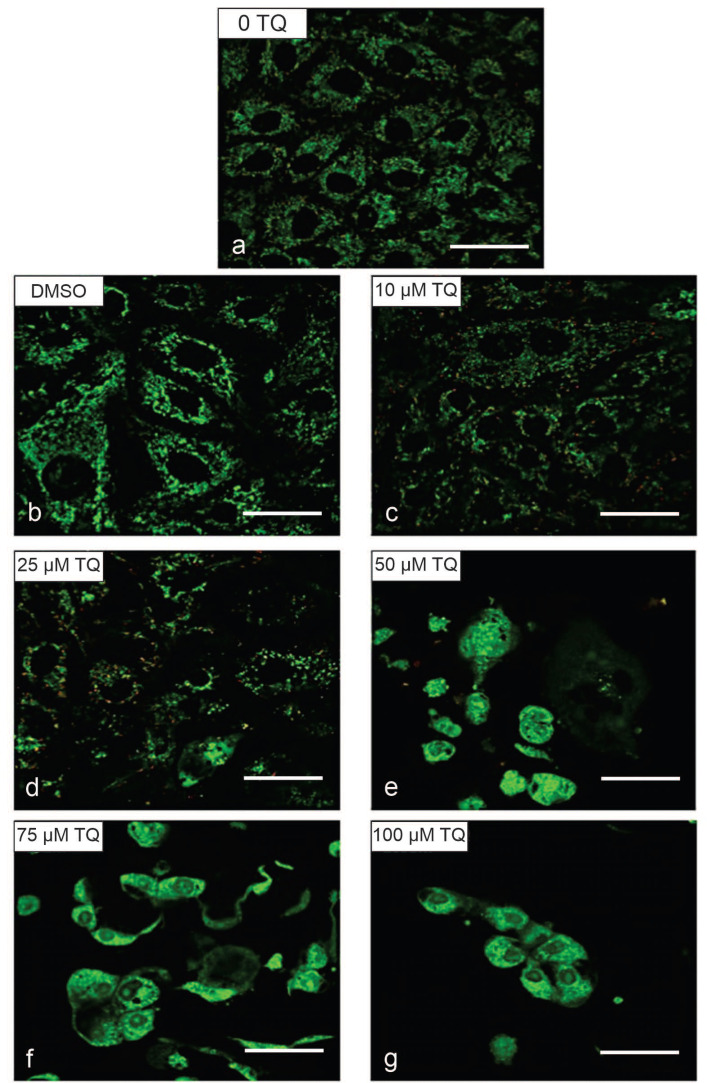
MMP disruption in HuCCT1 cells following exposure to TQ for 24 h (scale bar: 25 μm) MMP was evaluated using JC-1 by image techniques in the cancer cells after being exposed to 0–100 μM TQ: 0 μM TQ (**a**) and DMSO (**b**), and 10 (**c**), 25 (**d**), 50 (**e**), 75 (**f**), and 100 μM (**g**) TQ. The best representative images of each group are shown.

**Figure 11 ijms-23-14669-f011:**
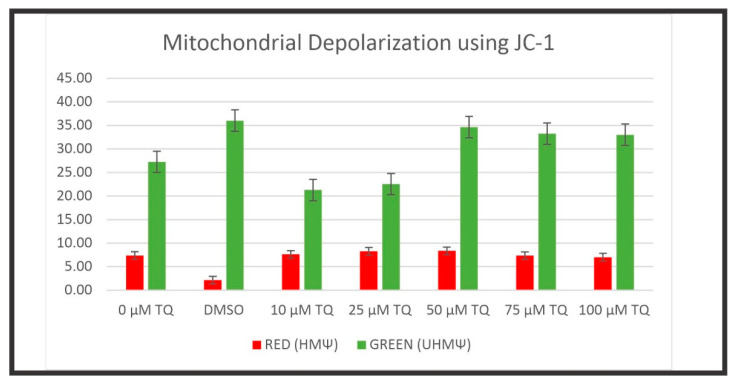
Mitochondrial depolarization using JC-1 for HuCCT1 using image analysis is demonstrated for DMSO and several concentrations of TQ. Three independent experiments were carried out for each concentration. A decrease in the red (~590 nm)/green (~529 nm) fluorescence intensity ratio after exposure to TQ is indicative of depolarization/disruption of the mitochondrial membrane. Three independent experiments were carried out for each concentration. Red means healthy mitochondria (HMΨ) while green means unhealthy mitochondria (UHMΨ). The variation in the mitochondrial depolarization is significant (*p* < 0.001, two-tailed test).

**Figure 12 ijms-23-14669-f012:**
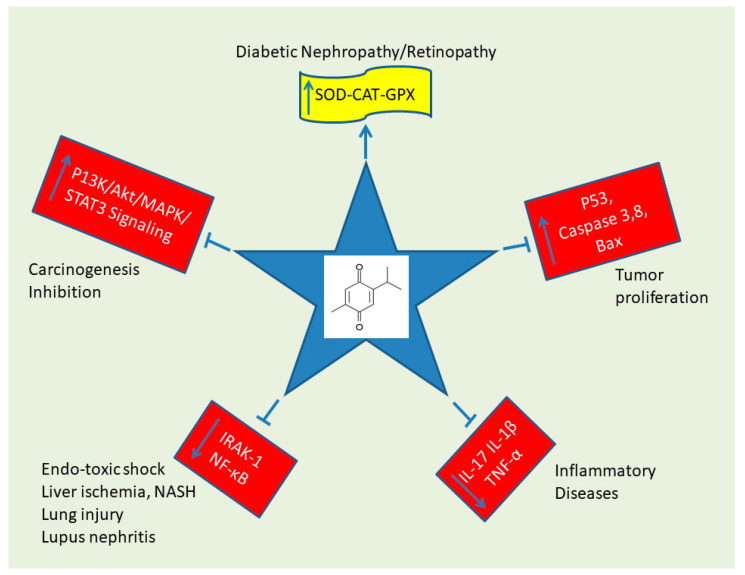
Thymoquinone Effects. See text for details about the various effects of TQ on the physiology and pathology.

**Table 1 ijms-23-14669-t001:** Constituents of the Volatile Oil of Black Seeds.

Component	Chemistry (IUPAC)	% of Volatile Oil
Thymoquinone (TQ)	2-Isopropyl-5-methylbenzo-1,4-quinone	28–57
ρ-cymene	1-Methyl-4-(propan-2-yl)benzene	7.1–15.5
Carvacrol (cymophenol)	2-Methyl-5-(propan-2-yl)phenol	5.8–11.6
4-Terpineol	2-(4-Methyl-1-cyclohex-3-enyl)propan-2-ol	2.0–6.6
Longifolene	4,8,8-trimethyl-9-methylidenedecahydro-1,4-methanoazulene	1.0–8.0
T-Anethole	1-Methoxy-4-[(1E)-prop-1-en-1-yl]benzene	0.25–2.3

IUPAC, International Union of Pure and Applied Chemistry.

**Table 2 ijms-23-14669-t002:** Cell Cycle Distribution of Cell Lines following Exposure to TQ treatments. Results shown are one representative of three independent experiments.

Cell Lines	Cell Cycle Phases	Control Group (%)	TQ-Treated Group (%)				
		0 TQ	10 μM	25 μM	50 μM	75 μM	100 μM
**HepG2**	**G1**	54.3	53.1	48.7	45.1	23.7	18.4
	**S**	20.0	20.4	18.2	16.8	13.3	9.3
	**G2/M**	22.6	23.4	28.4	23.1	14.5	6.8
	**SubG1**	2.6	2.4	2.8	18.1	54.1	60.7
**HuCCT1**	**G1**	46.8	34.9	51.9	56.4	58.2	60.3
	**S**	16.9	12.5	13.1	10.5	8.1	5.6
	**G2/M**	31.7	15.3	28.0	19.9	11.8	9.5
	**SubG1**	3.8	3.2	5.1	11.0	18.7	22.3

**Table 3 ijms-23-14669-t003:** The IC_50_ values of TQ in different cell lines after 12, 24, and 48 h of treatment.

Cell Lines	12 h	24 h	48 h
**HepG2**	50.74	34.23	50.84
**HuCCT1**	155.1	63.12	61.59

**Table 4 ijms-23-14669-t004:** Rodent Models targeting the Therapeutic Potential of TQ.

Model	Ref#	Carrier	TQ Dosage	Parameters	Mechanism	Potential
Mouse	[63]	APAP	0.5–2 mg/kg	ALT, NO_2_^−^/NO_3_^−^ LPX, GSH, ATP	↑AOE=>↓OXS, NO_2_^−^/NO_3_^−^ stress	Hep-Prot.
Mouse	[65]	DXR	10–20 mg/kg	AOE, MNM, LPO	↑AOE=>↓OXS, ↓ Inflammation	Cardio-Prot.
Mouse	[68]	TPA	1–5 μM	↑NFκB and COX-2	↓NFκB=>↓COX2	Anti-Infl.
Rats	[69]	Collagen	5 mg/kg	AOE and IMs	↑AOE=>↓OXS	Anti-Infl.
Rat	[62]	NaF	10 mg/kg	LFT, AOE	↑AOE=>↓OXS	Hep-Prot.
Rat	[92]	CCl_4_	5 mg/kg	ALT, GSSG, AO-mRNAs	↑AOE=>↓OXS	Hep-Prot.
Rats	[64]	IPN	20 mg/kg	OSM, MNM	Mitigation OXS, ↑AOE, ↑Tissue integrity	Cardio-Prot.
Rats	[66]	DXR	10 mg/kg	AOE, HSP70, HSP90, GRP78, caspase-3, ANP, NT-proBNP	↑AOE=>↓OXS, ↑ tissue integrity	Cardio-Prot.
Rats	[67]	AM	2.5–10 mg/kg	Gait score, MDA, GSH	↑AOE	Neuro-Prot.
Rats	[70]	STZ	80 mg/kg	Glu, TC, TG, HBA1c, MDA, TAC, Glut2 mRNA	↑ Glut-3 and IR	Anti-DM
Rats	[71]	STZ+NA	10 mg/kg	Glu, Lipids, HBA1c	Unclear	Anti-DM
Rats	[72]	STZ+NA	80 mg/kg	Glu, insulin, GSH, CAT, SOD, GPX, GST, LPO	↑AOE=>↓OXS	Anti-DM
Rats	[73]	FLS	0–10 μM	IMs and MP13	↓ NFκB and MAPK signaling pathway	AI-Prot.

**Notes**: ALT, alanine aminotransferase; AM, acrylamide; ANP, atrial natriuretic peptide; AOE, anti-oxidant enzymes; APAP, N-acetyl-para-aminophenol, paracetamol, acetaminophen; AST, aspartate aminotransferase; ATP, adenosin-triphosphate; CAT, catalase; CCl_4_, carbon tetrachloride; COX-2, cyclooxygenase 2; CPK, creatine phosphokinase; Glut2, glucose transporter 2; Glut-3, glucose transporter 3; FLS, fibroblast-like synoviocytes; GPX, glutathione peroxidase; GSH, glutathione; GSSG, glutathione disulfide; HBA1c, glycated hemoglobin; IMs inflammatory markers (TNF-α, IFN-γ, PGE2, IL-β, COX-2); IPN, isoprenaline or isoproterenol; IR, insulin receptor; LFTs, liver function tests; LPO, lipid peroxidation; MDA, malondialdehyde; MP13, metalloproteinase 13; MAPK, mitogen-activated protein kinase; MNM, myocardial necrosis markers (CPK, CK-MB, LDH, ALT, AST); NA, nicotinamide; NaF, sodium fluoride; NFκB, nuclear factor kappa-light-chain-enhancer of activated B cells; NO_2_^−^/NO_3_^−^, nitrites and nitrates; NT-proBNP, N-terminal (NT)-pro hormone b-type natriuretic peptide; OSMs, oxidative stress markers (GSH, MDA, and NO); OXS, oxidative stress; SOD, superoxide dismutase; STZ, streptozotocin; TAO, total antioxidant capacity; TC, total cholesterol; TGs, triglycerides. TAC is an analyte frequently used to assess the antioxidant status of biological samples and can evaluate the antioxidant response against the free radicals produced in each disease. TPA, 12-O-tetracanoylphorbol-13-acetate is a diester of phorbol. Trolox equivalent antioxidant capacity (TEAC), ferric-reducing ability of plasma (FRAP), and cupric-reducing antioxidant capacity (CUPRAC) are different assays described to determine TAC of a sample. GRP-78 or binding immunoglobulin protein (BiP) or heat shock 70 kDa protein 5 (HSPA5) is a protein that is encoded by the *HSPA5* gene in humans. AI-Prot., protective against autoimmune conditions or immunological dysregulations; Anti-DM, anti-diabetogenic; Anti-Infl., anti-inflammatory; Cardio-Prot., cardio-protective; Hep-Prot., hepato-protective; Neuro-Prot., neuro-protective. ↑AOE=>↓OXS means that an increase of antioxidant enzymes determines a decrease of oxidative stress.

## Data Availability

All essential data are provided in the manuscript. Additional data may be provided by the senior author (CMS) following reasonable requests.

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
