# Peer review of "Mitochondrial Dysfunction and Induction of Apoptosis in Hepatocellular Carcinoma and Cholangiocarcinoma Cell Lines by Thymoquinone"

_ijms, 2022, doi:10.3390/ijms232314669_

Round 1

Reviewer 1 Report

The findings appear to be interesting and specific points that the authors need to address are as follows:

1.  The manuscript lacks significant novelty as number of prior reports have already evaluated apoptotic effects of thymoquinone.

2. The mechanism by which thymoquinone can affect apoptosis should be investigated? Does thymoquinone also affect metastasis and invasion of tumor cells?

3. Proper statistical analyses should be done for all the data shown in the manuscript.

4. Acute toxicity studies should be done to establish the safety of thymoquinone?

5. The authors should provide their own justification and relevance of the study. This will help the readers to understand the importance of the paper.

6. The literature review in introduction should be improved by discussing important review articles published in the field.

Author Response

Thank you for the opportunity to receive your comments and suggestions. 

1. The manuscript lacks significant novelty as number of prior reports have already evaluated apoptotic effects of thymoquinone.

Indeed, there are early reports targeting the apoptotic effects of thymoquinone, However, this study is extensive and consolidates such effects. In particular, considering that some of the publications, such as Banerjee et al. in Cancer Res was retracted. That said, scientific reports are key in promoting research because they may support each other and discuss the evidence. As Prof. Arnqvist, of the Uppsala University, Sweden, indicated, "an assessment of novelty critically depends upon a reader's knowledge and perspective, the degree of novelty is arguably more demanding than many other criteria that can be used in editorial assessments." Also, the novelty criterion may lead many authors desperately striving to delineate the unique corner of a scientific territory, and the novelty criterion tends to reward poor scientific practice, because very little in science is fundamentally novel. The progress of science is built upon the scientific foundation formed by previous work creating a cornerstone of sound scientific practice. Substantially, papers attract readers because consolidate evidence or argue data. In addition, we revised the topic using a unique systematic review on rodent models that can be a fantastic platform for further studies in this direction. Thus, we humbly ask, and we would be very pleased if the reviewer could consider this argument.

2) The mechanism by which thymoquinone can affect apoptosis should be investigated? Does thymoquinone also affect metastasis and invasion of tumor cells?

Our manuscript indeed supports the evidence that thymoquinone affect apoptosis and the most recent paper of Jehan et al. identified a synergistic effect depending on the p53 status (Jehan et al. Front Pharmacol 2020). With regard to the metastasis and invasion of tumor cells, the systematic review on animal models indicates the efficacy to use this compound for therapeutical purposes. 

3) Proper statistical analyses should be done for all the data shown in the manuscript.

All statistical data have been certified by the Department of Statistics and Epidemiology of the University of Alberta. Personally, I cooperate with IARC/WHO on the carcinogenicity of several compounds and I fully agree that statistical data need to be fully verified by epidemiologists or statisticians.

4) Acute toxicity studies should be done to establish the safety of thymoquinone? 

Some toxicity data are presented in the manuscript, but extensive data will require animal models. That is the reason we added a systematic review on animal models.

5) The authors should provide their own justification and relevance of the study. This will help the readers to understand the importance of the paper.

We expanded the justification and the relevance of the study adding more references and some controversial publications.

6) The literature review in introduction should be improved by discussing important review articles published in the field.

We added more reviews published this field as requested.

We revised the English grammar and typing.

Reviewer 2 Report

The present investigates the TQ induced apoptosis against hepatocellular carcinoma (HepG2) and cholangiocarcinoma (HuCCT1) cells. The present study confirms that TQ caused cell cycle arrest at different phases and induced apoptosis in both cell lines. Although conducted experiments support the conclusion drawn, this study lacks novelty and the presentation of data needs to be improved to a great extent. An example is Figs’ 8 and 9 where authors present microscopic data but write it as flow cytometry data. The apoptotic effects of TQ have already been tested in HepG2 and other HCC line in this published article “Thymoquinone Selectively Induces Hepatocellular Carcinoma Cell Apoptosis in Synergism with Clinical Therapeutics and Dependence of p53 Status” doi: 10.3389/fphar.2020.555283. Unfortunately, I cannot recommend this article for publication.

Author Response

Thank you for the opportunity to receive your comments and suggestions. 

The present investigates the TQ induced apoptosis against hepatocellular carcinoma (HepG2) and cholangiocarcinoma (HuCCT1) cells. The present study confirms that TQ caused cell cycle arrest at different phases and induced apoptosis in both cell lines. Although conducted experiments support the conclusion drawn, this study lacks novelty and the presentation of data needs to be improved to a great extent. An example is Figs’ 8 and 9 where authors present microscopic data but write it as flow cytometry data. The apoptotic effects of TQ have already been tested in HepG2 and other HCC line in this published article “Thymoquinone Selectively Induces Hepatocellular Carcinoma Cell Apoptosis in Synergism with Clinical Therapeutics and Dependence of p53 Status” doi: 10.3389/fphar.2020.555283. Unfortunately, I cannot recommend this article for publication.

Indeed, there are early reports targeting the apoptotic effects of thymoquinone, However, this study is extensive and consolidates such effects. In particular, some of the publications, such as Banerjee et al. in Cancer Res was retracted. That said, scientific reports are key in promoting research because they may support each other and discuss the evidence. As Prof. Arnqvist, of the Uppsala University, Sweden, indicated, "an assessment of novelty critically depends upon a reader's knowledge and perspective, the degree of novelty is arguably more demanding than many other criteria that can be used in editorial assessments." Also, the novelty criterion may lead many authors desperately striving to delineate the unique corner of a scientific territory, and the novelty criterion tends to reward poor scientific practice, because very little in science is fundamentally novel. The progress of science is built upon the scientific foundation formed by previous work creating a cornerstone of sound scientific practice. Substantially, papers attract readers because consolidate evidence or argue data. In addition, we revised the topic using a unique systematic review on rodent models that can be a fantastic platform for further studies in this direction. Thus, we humbly ask, and we would be very pleased if the reviewer could consider this argument.

Our manuscript indeed supports the evidence that thymoquinone affect apoptosis and the most recent paper of Jehan et al. identified a synergistic effect depending on the p53 status (Jehan et al. Front Pharmacol 2020). With regard to the metastasis and invasion of tumor cells, the systematic review on animal models indicates the efficacy to use this compound for therapeutical purposes. 

All statistical data have been certified by the Department of Statistics and Epidemiology of the University of Alberta. Personally, I cooperate with IARC/WHO on the carcinogenicity of several compounds and I fully agree that statistical data need to be fully verified by epidemiologists or statisticians.

We revised the figures 8 and 9 as requested.

We would be very grateful if the reviewer may please consider our replies exposed as above.

We revised the English grammar and typing as requested. 

Round 2

Reviewer 1 Report

The authors have addressed all my concerns.

Author Response

Thank you for your comments and suggestions. We do appreciate your efforts to improve our manuscript.

Reviewer 2 Report

Although authors have provided justification for the novelty, the errors in the presentation of data still exists. Also, the quality of Fig. 7 is of poor with hard to read the numbers in the panels. I am mentioning the errors of legends in the red here.

As authors mention in the methods for JC1 that the dye fluoresces red/orange in intact and healthy mitochondria and fluoresces green in the cytoplasm in disrupted mitochondria, either a ratiometric quantification of red/green for the flow data must be shown, if the figure legends are correct, or if it was done using imaging technique then the legends must be corrected. 

Figure 8. MMP disruption in HepG2 cells following exposure to TQ for 24 hours (scale bar: 50 µm). MMP was measured using JC-1 by flow cytometry in the cancer cells after being exposed to 0 µM TQ, 0 µM TQ and DMSO, 10, 25, 50, 75, and 100 µM TQ. The best representative images of each group are shown.

Figure 9. MMP disruption in HuCCT1 cells following exposure to TQ for 24 hours (scale bar: 25 µm) MMP was measured using JC-1 by flow cytometry in the cancer cells after being exposed to 0 µM TQ, 0 µM TQ and DMSO, 10, 25, 50, 75, and 100 µM TQ. The best representative images of each group are shown. 

Author Response

Thank you for your comments and suggestions. We do appreciate your efforts to improve our manuscript.

Figure 7 has now a comprehensive legend highlighting the various concentrations of TQ. Figures 8 and 9 legends have been corrected and we added the image technique used to determine the mitochondrial depolarization using JC-1 (Figures 8-11).